# Detecting Embryo Developmental Potential by Single Blastomere RNA-Seq

**DOI:** 10.3390/genes14030569

**Published:** 2023-02-24

**Authors:** Monika Nõmm, Marilin Ivask, Pille Pärn, Ene Reimann, Sulev Kõks, Ülle Jaakma

**Affiliations:** 1Chair of Animal Breeding and Biotechnology, Institute of Veterinary Medicine and Animal Sciences, Estonian University of Life Sciences, Kreutzwaldi 62, 51014 Tartu, Estonia; 2Department of Pathophysiology, Institute of Biomedicine and Translational Medicine, University of Tartu, 50411 Tartu, Estonia; 3Competence Centre on Health Technologies, 50411 Tartu, Estonia; 4Estonian Genome Centre, Institute of Genomics, University of Tartu, 51010 Tartu, Estonia; 5Centre for Molecular Medicine and Innovative Therapeutics, Murdoch University, Perth, WA 6150, Australia; 6Perron Institute for Neurological and Translational Science, Perth, WA 6009, Australia

**Keywords:** single blastomere biopsy, single cell embryo biopsy, embryo developmental potential, preimplantation embryo selection, preimplantation embryo diagnostics

## Abstract

Recent advances in preimplantation embryo diagnostics enable a wide range of applications using single cell biopsy and molecular-based selection techniques without compromising embryo production. This study was conducted to develop a single cell embryo biopsy technique and gene expression analysis method with a very low input volume to ensure normal embryo development and to see if there are differences in gene expression profiles between day-5 biopsied bovine embryos that developed into blastocysts and embryos arrested at morula stage. Out of the 65 biopsied morulae, 32 developed to blastocysts (49.2%). Out of the 13,580 successfully annotated genes, 1204 showed a difference in mRNA expression level. Out of these, 155 genes were expressed in embryos developing to blastocysts. The pathway enrichment analysis revealed significant enrichment in “organelle biogenesis and maintenance”, “mRNA splicing” and “mitochondrial translation” pathways. These findings suggest principal differences in gene expression patterns and functional networks of embryos able to reach the blastocyst stage compared to embryos arrested in development. Our preliminary data suggest that single blastomere biopsy and selected gene expression profiles at morula stage could offer additional possibilities for early preimplantation embryo selection before transfer.

## 1. Introduction

Bovine embryo biopsy combined with molecular analysis has been a very useful preimplantation embryo assessment tool for over many decades [1,2,3,4,5]. It may be commercially used to determine the sex of the embryo, but also to select embryos for numerous genotypes for various production traits [6,7,8,9]. Analyzing bovine embryos at an early morula stage by removing only one blastomere lets the embryo regenerate and develop into a blastocyst at a normal rate and is less invasive than using a microblade or trophoderm biopsy [1,2,3,5,7,8,9,10,11,12,13]. In human medicine, early embryonic diagnostic testing is used to select the most viable embryos for transfer and screening for genetic diseases [10,11,12,13]. Bovine embryos have proven to be a good model for studying human in vitro produced (IVP) embryos because the bovine is a large mono-ovulating mammal and the embryo development kinetics is similar to that of humans. In addition, the preimplantation embryos seem to have similar biochemical and regulatory paternal and maternal processes [14,15,16]. For years, the main technique to select bovine IVP embryos for transfer has been visual morphological observation [1,2,4,5]. Observing only the morphology of the IVP embryos might not be accurate enough for predicting the embryo’s potential to implant and develop normally after transfer because unhealthy embryos can implant but are bound to miscarry during the first trimester of the pregnancy [1,2,4,5,10,11,12,13]. As the current assessment methods of embryo quality and development are not accurate enough, there is a need to develop an early minimally invasive embryonic screening system that can provide additional embryo quality information at an early stage of development before embryo transfer.

## 2. Aim

The aim of this study was to combine single blastomere biopsy with molecular-based analysis for bovine embryos at an early morula stage without impairing embryo development to detect differences in gene expression profiles of bovine IVP biopsied morulae that might give insight into the preimplantation bovine embryo gene expression and further development.

## 3. Materials and Methods

In the current study, slaughterhouse-derived material was used; therefore, no ethical approval was needed for the research. Unless otherwise stated, all chemicals were purchased from Sigma-Aldrich/Merck (Germany or the USA). The experimental design is illustrated in Figure 1.

### 3.1. In Vitro Embryo Production

Bovine embryos were made with slight modifications as previously described by [17]. Bovine ovaries (*Bos Taurus*) were transported from the slaughterhouse to the laboratory in 0.9% sterile NaCl solution within 4 h after slaughter at approximately 32–37 °C and washed twice in 0.9% NaCl solution. The cumulus oocyte complexes (COCs) were aspirated from follicles with a diameter of 2–8 mm. Quality code 1 [18] COCs were washed and matured in groups of 50 oocytes in 500 µL of in vitro maturation medium supplemented with 0.8% fatty acid-free fraction V bovine serum albumin (BSA) in four-well plates (Nuncoln/Thermo Fischer Scientific, Cat. No. 144444). Oocytes were incubated at 38.5 °C with 5% CO_2_ in humidified air for 22–24 h.

The matured oocytes were fertilized with frozen–thawed semen. Washed sperm was diluted to a final concentration of 2 × 10^6^ motile sperm per ml. Oocytes and sperm were co-incubated in groups of 50 in 500 µL of in vitro fertilization media in four-well plates at 38.5 °C with 5% CO_2_ in humidified air for 18–20 h.

Cumulus cells were removed from the presumptive zygotes by vortexing, and the denuded embryos were transferred into 500 µL of modified Synthetic Oviduct Fluid with amino acids and myo-inositol (SOFaaci) containing 0.8% BSA under mineral oil. The embryos were cultured at 38.5 °C, 5% CO_2_ and 90% N_2_ with humidified air for five days.

### 3.2. Embryo Biopsy and cDNA Synthesis

On day five, 65 morulae were biopsied with a microneedle and one blastomere was aspirated. The biopsied morulae were transferred into 50 µL culture media droplets under mineral oil and further individually cultured until day 8, when blastocyst formation was recorded.

The aspirated blastomere was directly transferred into 4 µL of lysis buffer (0.45% NP40, 4.5 mM DTT, 0.18 U/µL SUPERase-In, 0.36 U/µL RNase Inhibitor) using a 1 µL micropipette, incubated for 90 sec at 70 °C and placed on ice. The cDNA was amplified using the Ovation RNA-Seq System V2 Kit (Nugen/Tecan, Cat. No. 7102-08) and standard protocols. The MinElute Reaction Cleanup Kit (Qiagen, Cat. No. 28004) and standard protocols were used for cDNA purification. The quantity of each cDNA sample was assessed with the Qubit DNA HS Assay Kit (Thermo Fischer Scientific, Cat. No. Q32854). The quality of cDNA was evaluated using the Agilent DNA 1000 Kit and the 2100 Bioanalyzer (Agilent, Cat. No. 5067-1504). The obtained cDNA concentrations varied between 6.42 and 23.6 ng/µL. Synthesized cDNA was stored at −20 °C until further applications.

### 3.3. WT RNA Sequencing

For sequencing, six biopsy samples from embryos arrested at morula stage and six biopsy samples from embryos developing to the blastocyst stage were randomly chosen by quality analysis with a 2100 Bioanalyzer (Agilent).

For library preparation, all (previously) synthesized cDNA was used as input for a 5500 SOLiD Fragment Library Core Kit (Life Technologies/Thermo Fisher Scientific, Cat. No. 4464412). For sequencing, the SOLiD 5500 Wildfire platform was applied together with paired-end sequencing chemistry (50 bp forward and 50 bp reverse).

### 3.4. Mapping Raw RNA-Seq Data

Raw reads (75 bp) were color-space-mapped to the bovine genome bosTau7 reference using the Maxmapper algorithm implemented in Lifescope software (Life Technologies/Thermo Fisher Scientific). Mapping to multiple locations was permitted. The quality threshold was set to 10, so that the mapping confidence was more than 90. Reads with scores less than 10 were filtered out. The average mapping quality was 30. Analysis of the RNA contents and gene-based annotation were performed within a whole transcriptome workflow.

### 3.5. RNA-Seq Statistical Analysis

For statistical analysis of the RNA-seq data, the DeSeq2 package for R was used [19], which is specifically developed for RNA-seq or other count data. The package DeSeq2 provides methods to test for differential expression by the use of negative binomial distributions and a shrinkage estimator for a distribution’s variance [20]. The package performs sample comparisons and also adjusts *p*-values to overcome multiple testing problems. The DeSeq2 package uses the Benjamini–Hochberg procedure, which controls for the false discovery rate (FDR) [21].

### 3.6. Functional Annotation of Differentially Expressed Genes

In order to find functional relations between differentially expressed genes, pathway enrichment analysis was applied. The Bioconductor package ReactomePA for reactome pathway analysis and visualization implemented in R was used [22]. Enrichment analysis is a widely used method to identify biological themes in the complex lists of differentially expressed genes. ReactomePA implements a hypergeometric model to assess whether the number of selected genes associated with the reactome pathway is larger than expected. The *p*-values are calculated based on the hypergeometric model [22]. After analysis, the results were visualized using enrichment mapping and category-gene-network tools [22].

### 3.7. Quantitative Real-Time PCR Validation

For confirmatory quantitative real-time PCR (qPCR), a biopsy was taken from day-5 morulae and the biopsied embryos and blastomeres were processed exactly as for sequencing (Materials and Methods, Section 3.2), so the results would be comparable.

The qPCR was performed using TaqMan Gene Expression Master Mix (Thermo Fisher Scientific, Cat. No. 4369016) and the ViiA 7 Real-Time PCR System (Applied Biosystems/Thermo Fisher Scientific). The following TaqMan Gene Expression Assays (Thermo Fisher Scientific) were used: *SARS* (Bt03213059_m1), *HSBP1* (Bt03276404_m1) and *ATP5G3* (Bt03221537_g1). *GADPH* (Bt03210913_g1) was used as a reference housekeeping gene for normalization of mRNA expression levels of *SARS*, *HSBP1* and *ATP5G3*. Every sample was run in four parallel reactions. Cycle threshold (Ct) values were used to calculate the relative gene expression levels using the comparative Ct (∆Ct) method (according to Applied Biosystems/Thermo Fisher Scientific).

### 3.8. Quantitative Real-Time PCR Statistical Analysis

Data from qPCR are presented as means of 2^−ΔCt^ ± SDs. The data for the studied genes were analyzed by unpaired *t*-tests with Welch’s correction using GraphPad Prism 6 software (GraphPad Software., Boston, MA, USA), and a *p*-value < 0.05 was considered significant.

## 4. Results

### 4.1. Transcriptome Analysis

Out of the 65 biopsied morulae, a total of 32 (49.2%) developed to blastocysts. Out of the 13,580 successfully annotated genes, 1204 showed a difference in mRNA expression level with a *p*-value < 0.05. Out of these, 155 genes had a log_2_FC value over 2 and were expressed in embryos developing into blastocysts after biopsy. The most expressed genes were seryl-tRNA synthetase (*SARS*), heat shock factor binding protein 1 (*HSBP1*), ATP synthase, H^+^ transporting, mitochondrial Fo complex subunit C3 (subunit 9) (*ATP5G3*), ubiquinone oxidoreductase complex assembly factor 5 (*NDUFAF5*, also known as *C13H20orf7)* and cystatin B (*CSTB*) (Figure 2, Table 1). Embryos arrested at morula stage after biopsy had 1836 genes with a log_2_FC value over 2. The most expressed genes were chemokine (C-X-C motif) receptor 4 (*CXCR4*), exostosin glycosyltransferase 1 (*EXT1*), cerebellar degeneration related protein 2 (*CDR2*), synuclein α (*SNCA*) and mediator complex subunit 27 (*MED27*) (Figure 2, Table 2). The pathway enrichment analysis revealed significant enrichment in “organelle biogenesis and maintenance”, “mRNA splicing” and “mitochondrial translation” pathways, with a *p*-value < 0.015 (Figure 3) between the two biopsied embryo groups. Principal Component Analysis (PCA) was performed using rlog-transformed data with DeSeq2 built-in plotPCA (Figure 4). Components 1 (x-axis) and 2 (y-axis) separated the normally developing embryos that reached the blastocyst stage and embryos, which development was arrested at morula stage. The samples showed no grouping by embryo development in the first two dimensions (Figure 4).

The results of the transcriptome analysis were also visualized on an MAplot which shows the log2 fold changes attributable to a given variable over the mean of normalized counts for all the samples in the DeSeq DataSet. Points that are colored red differ significantly with the adjusted *p*-value less than 0.1 (Figure 5).

### 4.2. Quantitative Real-Time PCR Confirmation

To determine whether changes in gene expression similar to those observed with RNA-seq could be observed with qPCR, the expression patterns of *SARS*, *HSBP1* and *ATP5G3* were analyzed in both sample groups.

According to the RNA-seq results, all the selected genes were more expressed in embryos developing into blastocysts. With qPCR, only the upregulation of *ATP5G3* in blastocysts was statistically confirmed (*p*-value = 0.0366) (Figure 6). Although *SARS* showed a trend for a higher expression in embryos developing to blastocysts, this was not statistically confirmed due to high variation (Figure 6). *HSBP1*, on the contrary, was more expressed in embryos arrested in development according to qPCR, but this was not statistically confirmed (Figure 6).

## 5. Discussion

Previous studies have shown that embryo biopsy is a promising and valuable tool for preimplantation diagnostics for farmers and research purposes [5,23,24,25]. In this study, the first aim was to develop a single blastomere biopsy technique for embryos at an early morula stage that allows the application of different molecular-based selection techniques with a low input volume, while at the same time being as minimally invasive as possible to ensure normal embryo development without compromising embryo production. The second aim was to determine whether there are differences in gene expression patterns between day-5 biopsied bovine embryos that developed into blastocysts and embryos with arrested development at morula stage after biopsy.

The results showed that removing one blastomere at an early morula stage does not alter embryo development and that 49.2% of the biopsied morulae developed further into blastocysts. Previous studies showed similar results for single embryo culture systems, with blastocyst rates of 33.5% and 39.3%, respectively, without biopsy [26,27]. Our results agree with other studies where one or multiple blastomeres were removed from an early-morula-stage embryo with blastocyst rates of 37.1%, 57.5% and 53.3% [5,25,28], respectively. With this study we demonstrated that removing only one blastomere at an early morula stage can provide enough material for RNA sequencing and quantitative real-time PCR. Tutt et al. (2020) presented similar results using laser-assisted blastomere removal, showing that removing one to three blastomeres from one morula does not alter embryonic development [25]. However, using a laser might affect sample quality for further analysis; thus, some samples in the PCR analyses resulted in a number of false positives [25]. Polisseni et al. (2010) performed whole genome amplification and PCR analyses for sex determination of bovine embryos, the number of blastomeres removed from each embryo was equivalent to a quarter of the embryo, which did not alter blastocyst formation [5].

In human in vitro embryo production, preimplantation genetic testing (PGT) is a common practice, especially for patients with a high risk of a genetic diseases [29,30,31]. For example, embryos might be screened for aneuploidy, which is the presence of an abnormal number of chromosomes in a cell, by analyzing biopsies of day-3 or day-4 morulae. This allows the selection of a chromosomally normal embryo for transfer for the development of healthy offspring [29,30,31]. Therefore, good and minimally invasive molecular preimplantation testing methods and practices are essential in assisted reproduction.

After biopsy and RNA sequencing, the embryos that developed into blastocysts had higher expression of the genes *SARS*, *HSBP1*, *ATP5G3*, *C13H20orf7* and *CSTB* (Figure 2). *HSBP1* belongs to the heat shock protein (HSP) group, which might be markers for preimplantation embryo development [32]. *SARS* and *ATP5G3* are both involved in reprogramming mitochondrial metabolism [33] and participate in the biosynthesis of selenocysteine in the mitochondria. Our findings support the understanding that proper mitochondrial function is crucial for embryonic development and sustaining normal preimplantation embryonic growth [33]. Mitochondria are known as “the powerhouse of the cell” by generating ATP through oxidative phosphorylation, thus providing energy for the cell to grow [34,35,36,37,38]. Mitochondria are maternally inherited and are the most abundant organelles in the oocyte and preimplantation embryo [34,35,36,37]. During embryo development, mitochondria undergo major structural and positional changes, but their number (~100,000) in the embryo remains constant [34,35,36,37,38]. Therefore, the more mitochondria there are in an oocyte originally, the better the quality of the oocyte and the higher the probability of normal fertilization and development into a blastocyst [34,35,36,37,38]. Furthermore, in the current study, the pathway enrichment analysis revealed significant enrichment in “organelle biogenesis and maintenance”, “mRNA splicing” and “mitochondrial translation” pathways (Figure 3), supporting the importance of proper mitochondrial function for the development of an embryo. Although the expression of the *HSBP1* gene was not statistically confirmed with the results of RNA sequencing and qPCR analysis in the current study, it has been shown earlier that heat shock proteins (HSP) play a fundamental role in livestock gamete formation and embryonic development. Driver and Khatib (2013) analyzed different HSPs and their impacts on in vitro and in vivo embryo development [32]. According to their results, HSPs maintain normal cell development and address stress, but, interestingly, the HSP40 family is upregulated in degenerated embryos [32]. In the current study, based on qPCR results, *HSBP1* was more expressed in embryos arrested at morula stage, supporting previous findings. Therefore, the presence of HSP should further be characterized throughout early embryo development, especially when IVP bovine blastocysts tend to be morphologically similar, and selecting the best-quality embryos for transfer is essential [32].

The cluster analysis showed distinct differences in gene expression patterns between the blastocyst group and the embryos arrested in development at morula stage (Figure 2). Some of the genes represented in the morula group, for example, *SMARCA2*, *CASP2* and *TOB1***,** are related to impaired embryonic development, cell death/apoptosis and embryonic lethality [39,40,41]. On the other hand, in the group that developed into blastocysts, genes such as *SIRT1*, *SIRT7* and *OLA1* (Figure 2) were represented. Sirtuin-1 (*SIRT1*) is known as a regulator of mitochondrial biogenesis and turnover [42]. *SIRT1* might maintain mitochondrial quality in the embryo and therefore improve its development by clearing out damaged mitochondria in a dying cell, so that these are not passed down to daughter cells during cell division [42]. In porcine embryos, *SIRT1* activation has proven to have a positive effect on the blastocyst formation rate [43]. Aksu et al. (2021) found that *SIRT1* may protect the embryo from oxidative stress during implantation through *FoxO1*-*SOD* signaling [44]. *SIRT7* is an essential factor in the determination of oocyte quality, as *SIRT7*-depleted oocytes generate aneuploid eggs and their reactive oxygen levels are elevated, thereby compromising the developmental competence of the oocytes [45]. In mouse embryos, the lack of *Ola1* reduced the growth of the embryos and led to immature organ formation and prenatal lethality [46]. Thus, our findings are in good agreement with earlier reports showing distinct gene expression patterns between embryos arrested in development and those that developed into blastocysts.

The differences in early embryonic gene expression are crucial for normal blastocyst development and may provide additional opportunities for better embryo selection. Studies on the human preimplantation embryo biopsy suggests the same, although these studies mostly focus on trophoderm biopsy, which is a more common biopsy method used in preimplantation embryos [47,48,49]. There are also some studies that have been profiling different stages of preimplantation human and mouse embryos by sequencing, but these results differ from ours [50,51,52]. In these studies, the authors have used whole embryos or trophoderm biopsy, which is different from our experimental design, and therefore the observed gene expression profiles were different from our present results [50,51,52]. Trophoderm biopsy is a good tool to predict blastocyst implantation and does not affect embryonic development, as it is taken in the blastocyst stage [47,49]. Using trophoderm biopsy and RNA-seq, it is possible to identify preimplantation embryo sex, karyotype and candidate gene sets for developmentally competent embryos [47,49]. These results show that embryo biopsy and RNA-seq have future potential for clinical application in human assisted reproduction and also in cattle IVP embryos for breeding purposes [47].

### Study Limitations

As not all of the selected genes could be statistically confirmed with qPCR, the study could have benefited from analyzing more samples with the selected genes, as the variation among individual samples with qPCR is high. In addition, transfer studies are needed to confirm the biological relevance of the different transcriptome profiles for having live offspring.

Although there are limitations to the study and the data might not yet be sufficiently confirmative, it provides material and a basis for future studies.

## 6. Conclusions

With the increasing need for faster and more accurate preimplantation embryo development potential evaluation methods, single blastomere analysis at the morula stage could offer additional information for early embryonic diagnostics. We demonstrated that single blastomere RNA sequencing could provide insight into the preimplantation embryo, which could be applied to early preimplantation embryo selection via qPCR analysis. The current findings suggest that there are principal differences in the gene expression patterns and functional networks between developing embryos that are able to reach the blastocyst stage and embryos arrested in development. Single blastomere biopsy at the morula stage allows embryo analysis without compromising embryo production. Further studies are necessary to confirm whether single cell biopsy at the morula stage and gene expression analysis could be useful as complementary methods for non-invasive embryo selection before transfer. These findings should be further validated by embryo transfers and live births.

## Figures and Tables

**Figure 1 genes-14-00569-f001:**
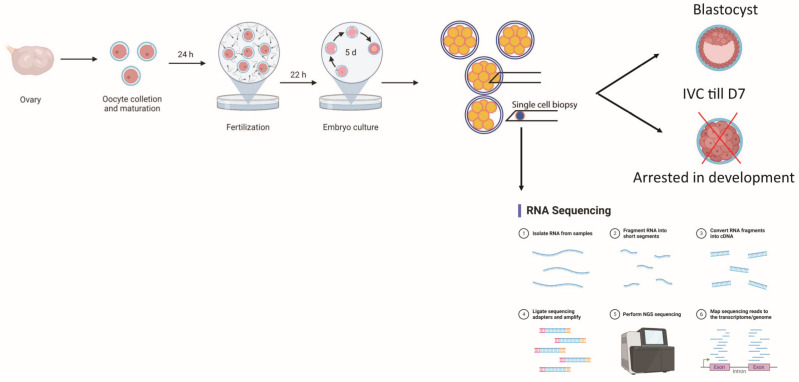
Summary of the experimental design.

**Figure 2 genes-14-00569-f002:**
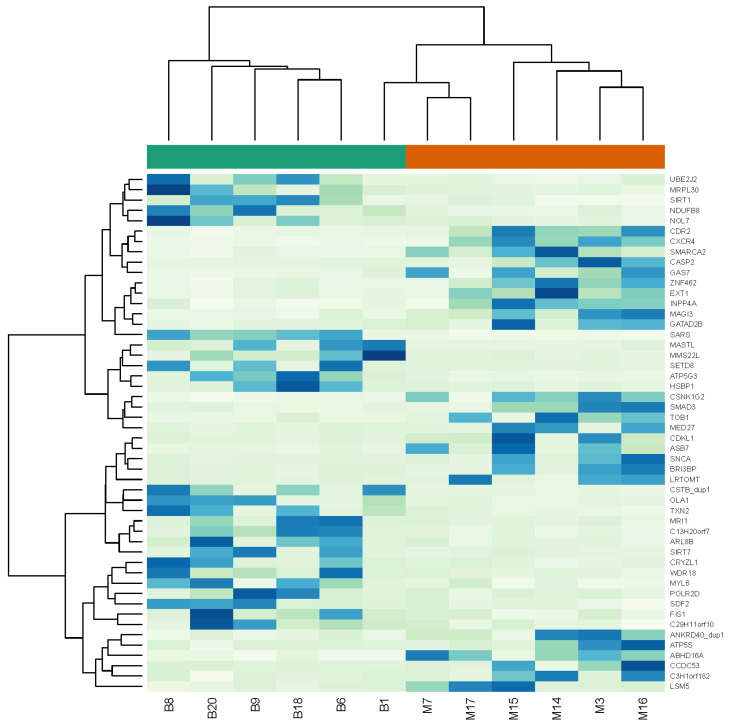
Heat map and cluster dendrogram of the 50 most differentially expressed genes. The scaled expression of each probe set, denoted as the row Z-score, is plotted on a green–blue color scale, with green indicating low expression and blue indicating high expression. Green (B)—embryos developed into blastocysts after biopsy; red (M)—embryos arrested as morulae after biopsy.

**Figure 3 genes-14-00569-f003:**
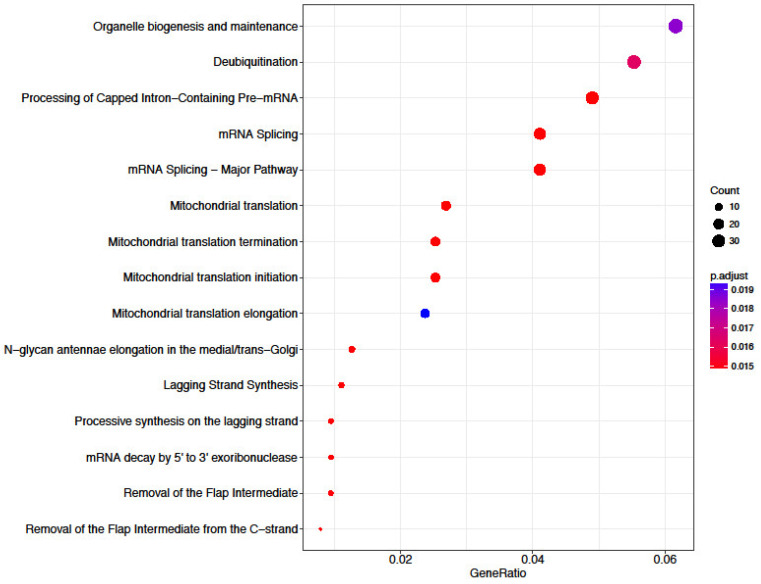
Pathway dot plot of gene expression profiles in between the biopsied embryo groups.

**Figure 4 genes-14-00569-f004:**
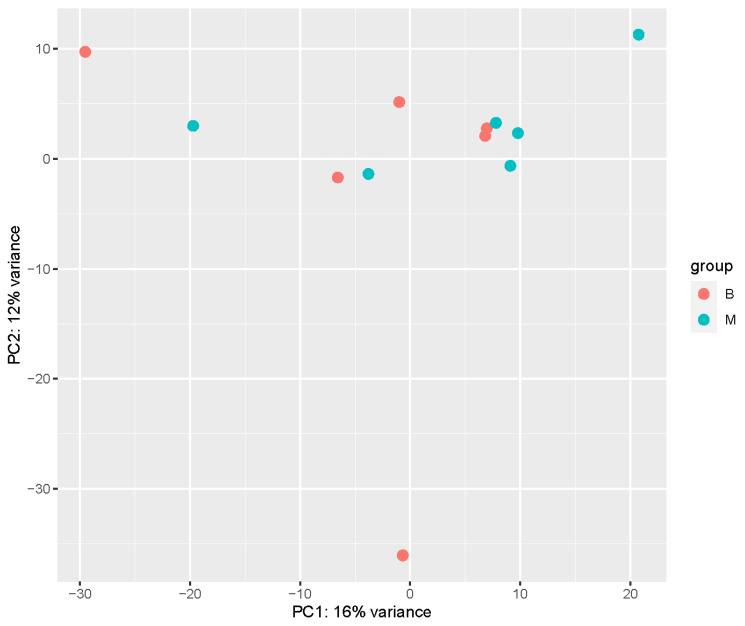
PCA plot. In red (B), embryos that developed to blastocysts after biopsy; in blue (M), embryos which development was arrested at morula stage.

**Figure 5 genes-14-00569-f005:**
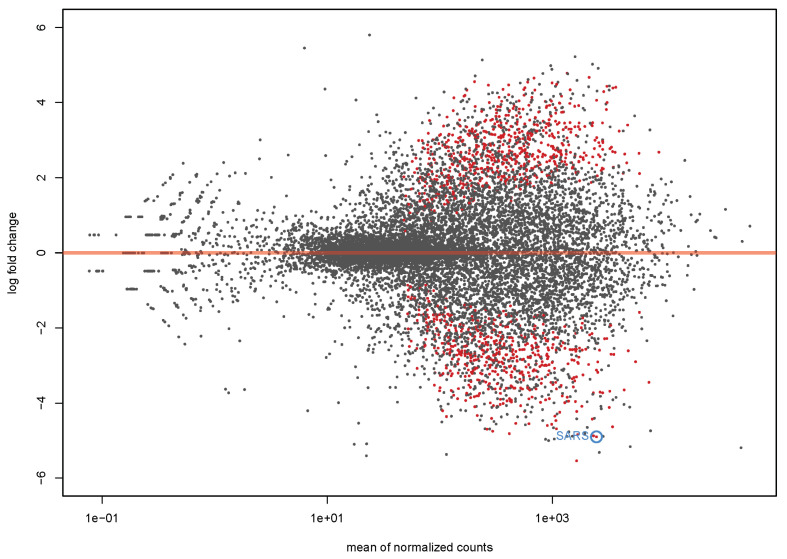
MAplot. The log2 fold changes attributable to a given variable are shown over the mean of normalized counts for all the samples in the dataset. Points that are colored red differ significantly with the adjusted *p*-value less than 0.1.

**Figure 6 genes-14-00569-f006:**
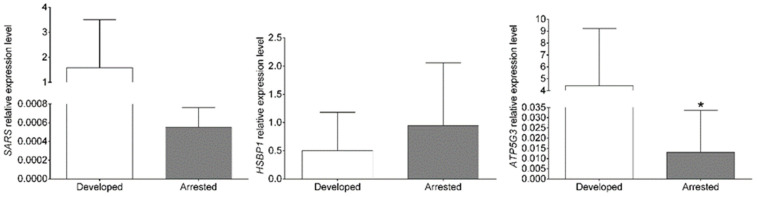
Confirmatory gene expression analysis with qPCR. Statistical analysis by unpaired *t*-tests with Welch’s correction, where * *p*-value < 0.05. Data plotted as means ± SDs, *n* = 4.

**Table 1 genes-14-00569-t001:** Top 10 genes expressed in embryos developing into blastocysts.

Symbol	Gene Name	log2 Fold Change	*p*-Value
*SARS*	seryl-tRNA synthetase	−4.25	6.81 × 10^−14^
*HSBP1*	heat shock factor binding protein 1	−3.87	3.42 × 10^−7^
*ATP5G3*	ATP synthase, H^+^ transporting, mitochondrial Fo complex subunit C3 (subunit 9)	−3.77	2.72 × 10^−9^
*C13H20orf7*	NA	−3.74	4.18 × 10^−8^
*CSTB_dup1*	NA	−3.65	3.83 × 10^−8^
*MRPL30*	mitochondrial ribosomal protein L30	−3.61	NA
*MMS22L*	MMS22 like, DNA repair protein	−3.57	NA
*MRI1*	methylthioribose-1-phosphate isomerase 1	−3.56	1.04 × 10^−6^
*UCHL3_dup2*	NA	−3.55	NA
*NDUFB8*	NADH:ubiquinone oxidoreductase subunit B8	−3.52	1.97 × 10^−7^

**Table 2 genes-14-00569-t002:** Top 10 genes expressed in embryos arrested at morula stage.

Symbol	Gene Name	log2 Fold Change	*p*-Value
*CDKL1*	cyclin dependent kinase like 1	3.31	4.37 × 10^−6^
*GATAD2B*	GATA zinc finger domain containing 2B	3.32	3.63 × 10^−6^
*LSM5*	LSM5 homolog, U6 small nuclear RNA and mRNA degradation associated	3.32	7.66 × 10^−7^
*C14H8orf59*	chromosome 14 open reading frame, human C8orf59	3.33	NA
*BRI3BP*	BRI3 binding protein	3.34	4.02 × 10^−6^
*MED27*	mediator complex subunit 27	3.37	3.03 × 10^−6^
*SNCA*	synuclein α	3.40	2.76 × 10^−6^
*CDR2*	cerebellar degeneration related protein 2	3.40	9.00 × 10^−9^
*EXT1*	exostosin glycosyltransferase 1	3.46	NA
*CXCR4*	chemokine (C-X-C motif) receptor 4	4.10	2.34 × 10^−12^

## Data Availability

Data available upon personal request.

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
