# Peer review of "Detecting Embryo Developmental Potential by Single Blastomere RNA-Seq"

_genes, 2023, doi:10.3390/genes14030569_

Round 1

Reviewer 1 Report

The present manuscript investigates the possibility to use single cell biopsy in preimplantation embryos to identify differences in gene expression patterns between embryos developed into blastocyst and embryos with arrested development at morula stage. Furthermore the researchers develop a single blastomere biopsy technique, for the application of different molecular selection techniques with low input volume and without compromising embryo normal development. In my opinion, this manuscript looks interesting, there are however, few suggestions that could enrich the manuscript and I think are important.

Materials and Methods:

-          All the used products and reactors should be completed with the reference together with the commercial house. This is important for the reproducibility os the experiments.

-          In my opinion there is a lack of description of the bioinformatics analysis of the single cell RNA-Seq. The authors should include all the DeSeq2 previous and later steps, the image of the PCA plots, some alignment statistics etc at least in supplementary material.

-          In 3.2 and 3.7 section, there is a description of RNA extraction and cDNA synthesis, which is more or less the same words. Is it possible to write both sections together in a section called RNA extraction, and then follow it with RNA sequencing or RT-PCR validation?

Results:  

-          When talking about the differentially expressed genes, authors describe both groups, first the one of the embryos developing into blastocyst after biopsy and second the embryos arrested at morula stage after biopsy. However, when talking about the pathway enrichment analysis the authors only contemplate the first group. In my opinion, it is also interesting to analyse the enrichment pathway of the arrested embryos.

-          In Figure 3 legend authors have include a large figure legend, which seems more a result. In my opinion they should shortened the legend and add explanations to results paragraph.

Discussion:

-          The authors work in bovine animal model. Do they know if there are similar results in other animal models? The do not discuss in this sense.

-          Is there any connection between the identified genes? Authors can use different tools such as genemania see if all identified genes have any pathway or mechanism in common.

-          Author explain in limitations that transfer studies are needed, in my opinion this next step is a really interesting thing to have in consideration to round out the work.

-          Which type of clinical application have these results? Authors should discuss it. Do authors think that these results could be extrapolated to humans and use them in clinic?

Author Response

The present manuscript investigates the possibility to use single cell biopsy in preimplantation embryos to identify differences in gene expression patterns between embryos developed into blastocyst and embryos with arrested development at morula stage. Furthermore the researchers develop a single blastomere biopsy technique, for the application of different molecular selection techniques with low input volume and without compromising embryo normal development. In my opinion, this manuscript looks interesting, there are however, few suggestions that could enrich the manuscript and I think are important.

Materials and Methods:

-          All the used products and reactors should be completed with the reference together with the commercial house. This is important for the reproducibility of the experiments.

We have added all the reference numbers to the used products and reagents.

-          In my opinion there is a lack of description of the bioinformatics analysis of the single cell RNA-Seq. The authors should include all the DeSeq2 previous and later steps, the image of the PCA plots, some alignment statistics etc at least in supplementary material.

We used standard methods described in the LifeScope mapping software and DeSeq2 package, however we have added additional details and a PCA plot to the manuscript.

-          In 3.2 and 3.7 section, there is a description of RNA extraction and cDNA synthesis, which is more or less the same words. Is it possible to write both sections together in a section called RNA extraction, and then follow it with RNA sequencing or RT-PCR validation?

We have modified the sections 3.2 and 3.7 to reduce the repetition.

Results: 

-          When talking about the differentially expressed genes, authors describe both groups, first the one of the embryos developing into blastocyst after biopsy and second the embryos arrested at morula stage after biopsy. However, when talking about the pathway enrichment analysis the authors only contemplate the first group. In my opinion, it is also interesting to analyse the enrichment pathway of the arrested embryos.

We thank the reviewer for pointing out that the pathway enrichment analysis for the arrested embryos would provide new and interesting data about the preimplantation embryo, unfortunately for this article the analysis was only done for the group of biopsied embryos that developed into blastocysts.

-          In Figure 3 legend authors have include a large figure legend, which seems more a result. In my opinion they should shortened the legend and add explanations to results paragraph.

We have modified the figure legend and added the explanations to the appropriate results paragraph.

Discussion:

-          The authors work in bovine animal model. Do they know if there are similar results in other animal models? The do not discuss in this sense.

Similar studies have been performed also in mouse models (Zeng et. al 2005) and in preimplantation human embryos (Petropoulos et al., 2016; Yan et al., 2013). However, they used the whole preimplantation embryo and we used only one biopsied blastomere. We have added the results in the discussion part of the manuscript.

-          Is there any connection between the identified genes? Authors can use different tools such as genemania see if all identified genes have any pathway or mechanism in common.

We thank the reviewer for this suggestion, but there is no reference genome for Bos Taurus in such tools, because this information is commercially very valuable for big breeding companies and therefore classified. In our study we do not know the exact interactions in the identified genes, but based on the pathway enrichment analysis of embryos developing to blastocysts revealed significant enrichment in “organelle biogenesis and maintenance”, “mRNA splicing” and “mitochondrial translation” pathways. This information is valuable for future interaction studies.

-          Author explain in limitations that transfer studies are needed, in my opinion this next step is a really interesting thing to have in consideration to round out the work.

Our future plans involve generating a panel of genes as preimplantation embryo quality biomarkers and test them in embryo transfer studies.

-          Which type of clinical application have these results? Authors should discuss it. Do authors think that these results could be extrapolated to humans and use them in clinic?

Adding gene expression analyses to the clinical routine would not be possible in large scale, although some authors have suggested it (Ntostis et. al 2019; Groff et. al 2019). We would suggest that the information based on our and other studies should be applied as embryo biopsy on day 5 after fertilization (for cattle, day 3 or 4 for human) and using a panel of genes that might act as biomarkers for preimplantation embryo developmental competence. This could be used as additional method for selecting the best quality embryos for transfer to insure live births.

Reviewer 2 Report

In this study the authors aimed to develop a single cell embryo biopsy technique and gene expression analysis method to ensure normal embryo development and to see if there are differences in gene expression profiles between day 5 biopsied bovine embryos that developed into blastocysts and embryos arrested at morula stage. Their data suggest that single blastomere biopsy and selected gene expression profiles at morula stage could offer additional possibilities for early preimplantation embryo selection before transfer. 

The manuscript is clear and well written. However, some methodologial issues should clarify to unsure an accurate study design. In particular:

1. The authors should further describe the technique used to perform morula biopsy on day 5. It is not common, as embryo at the cleavage stage and blastocyst in parlicular are usually biopsied. Please add a reference

2. The authors compared genes expression between morula developed to blastocyst and those who underwent developmental arrest. I think that a third group, comprising embryos  timely reaching the blastocyst stage on day 5, should add as control 

3. The authors suggested that this approach woud be used for embryo selection prior transfer. Please futher discuss a preliminar cost-effectiveness evaluation for the introduction of biopsy coupled to gene expression in the clinical routine of the IVF lab

Author Response

Reviewer 2

In this study the authors aimed to develop a single cell embryo biopsy technique and gene expression analysis method to ensure normal embryo development and to see if there are differences in gene expression profiles between day 5 biopsied bovine embryos that developed into blastocysts and embryos arrested at morula stage. Their data suggest that single blastomere biopsy and selected gene expression profiles at morula stage could offer additional possibilities for early preimplantation embryo selection before transfer.

The manuscript is clear and well written. However, some methodologial issues should clarify to unsure an accurate study design. In particular:

  1. The authors should further describe the technique used to perform morula biopsy on day 5. It is not common, as embryo at the cleavage stage and blastocyst in parlicular are usually biopsied. Please add a reference

Preimplantation embryo blastomere biopsy is a common practice at day 5 in cattle embryos for an example embryo sexing before transfer. In human IVP blastomere biopsy is used at day 3 or 4 after fertilization for aneuploidy testing. We have added references to the manuscript.

  1. The authors compared genes expression between morula developed to blastocyst and those who underwent developmental arrest. I think that a third group, comprising embryos timely reaching the blastocyst stage on day 5, should add as control

Although it is a great idea to add a third group as a control, we did not see any delay in embryo development after the biopsy. In bovine, normal blastocyst development is recorded on day 7 and 8 not on day 5 as it might be for the human IVF embryos.

  1. The authors suggested that this approach would be used for embryo selection prior transfer. Please futher discuss a preliminar cost-effectiveness evaluation for the introduction of biopsy coupled to gene expression in the clinical routine of the IVF lab.

Adding gene expression analyses to the clinical routine would not be possible in large scale, although some authors have suggested it (Ntostis et. al 2019; Groff et. al 2019). We would, however, suggest that the information based on our and other studies should be applied as embryo biopsy on day 5 after fertilization (for cattle, day 3 or 4 for human) and using a panel of genes that might act as biomarkers for preimplantation embryo developmental competence. This could be used as additional method for selecting the best quality embryos for transfer to insure live births, when these methods and protocols are fully standardized and certified.

Round 2

Reviewer 1 Report

-All the images have very low quality, so authors need to improve it.

-Some sentences seem to be unfinished, authors need to read the whole manuscript after the changes and improve the writing.

Materials and Methods:

-          In my opinion there is a lack of description of the bioinformatics analysis of the single cell RNA-Seq. The authors should include all the DeSeq2 previous and later steps, as a brief summary.

Results: 

-          When talking about the differentially expressed genes, authors describe both groups, first the one of the embryos developing into blastocyst after biopsy and second the embryos arrested at morula stage after biopsy. However, when talking about the pathway enrichment analysis the authors only contemplate the first group. I strongly believe, that it is also interesting to analyse the enrichment pathway of the arrested embryos.

-          In Figure 4, there is no explanation about (M) item. And in figure 5 there is no explanation of figure legend.

-          If both groups are not separated when doing the PCA plot, it is difficult to believe that observed gene changes after the comparison, are due to the different development that awaits them or due to chance.

Discussion:

-          In this last version, there is not the title of the discussion section.

-          The authors work in bovine animal model. Do they know if there are similar results in other animal models? I do not see were they have include this information in discussion.

-          Which type of clinical application have these results? Do authors think that these results could be extrapolated to humans and use them in clinic? Authors should extend the discussion in this aspect.

Author Response

Reviewer 1

-All the images have very low quality, so authors need to improve it.

We will add complementary images to the manuscript. The quality of the images is due to formatting.

-Some sentences seem to be unfinished, authors need to read the whole manuscript after the changes and improve the writing.

We will reread the manuscript and change the sentences accordingly.

Materials and Methods:

-          In my opinion there is a lack of description of the bioinformatics analysis of the single cell RNA-Seq. The authors should include all the DeSeq2 previous and later steps, as a brief summary.

After mapping with Lifescope, the bam files were imported to R using summarizeOverlaps function and read counts matrix was generated as part of a summarizedExperiment object. From here DESeqDataSet object was formed by using DESeqDataSet function with embryo outcome as a design element. For differential gene expression analysis DESeq function was used and results were annotated with the package org.Bt.eg.db.

Results:

-          When talking about the differentially expressed genes, authors describe both groups, first the one of the embryos developing into blastocyst after biopsy and second the embryos arrested at morula stage after biopsy. However, when talking about the pathway enrichment analysis the authors only contemplate the first group. I strongly believe, that it is also interesting to analyse the enrichment pathway of the arrested embryos.

Thank you for pointing out our miswording. The pathway enrichment is based on the list of the genes differentially expressed, it is comparative or relative. The list of genes is not only for BL or Morula, the list shows the difference between these two. And the pathway analysis shows the difference between these two as well and in this way it describes not only the pathways that are up-regulated in BL, but also pathways that are up-regulated in Morulas and also down-regulated in both groups. Heat-map gives very good overview of these genes.

-          In Figure 4, there is no explanation about (M) item. And in figure 5 there is no explanation of figure legend.

In figure 4 in blue (M) embryos which development arrested at morula stage. We have re-added the explanation of the figure legend to figure 5.

-          If both groups are not separated when doing the PCA plot, it is difficult to believe that observed gene changes after the comparison, are due to the different development that awaits them or due to chance.

We would like to explain that PCA plot shows how variable are the samples due to some general trends inside the sampling. PC1 and PC2 show the variability, so in this case PC1 describes 16% of all variability and PC2 12%.

PCA plot does not perform any formal statistical comparison, it is multidimensional statistical generalization that could indicate some bigger trends inside the datasets, but it never shows genes that are differentially expressed. The formal group wise comparison that is performed with DESeq2 is only way to get differential gene expressions between the groups.

Discussion:

-          In this last version, there is not the title of the discussion section.

We thank the reviewer for this remark, there was a layout error.

-          The authors work in bovine animal model. Do they know if there are similar results in other animal models? I do not see were they have include this information in discussion.

There are also some studies that have been profiling different stages of preimplantation human and mouse embryos by sequencing but these results differ from ours because the authors used whole embryos or trophoderm biopsy for RNA-sequencing (Petropoulos et al., 2016; Yan et al., 2013; Zeng & Schultz, 2005).

-          Which type of clinical application have these results? Do authors think that these results could be extrapolated to humans and use them in clinic? Authors should extend the discussion in this aspect.

We would argue with the reviewer as the results are yet preliminary and further studies are needed to confirm our results. There is indication for future use of these results in human clinics and we have discussed this aspect sufficiently.

Reviewer 2 Report

The authors have addressed my comments accordingly

Author Response

We thank the Reviewer 2 for the input and help to improve our manuscript.